# Analyzing Amylin Aggregation Inhibition Through Quantum Dot Fluorescence Imaging

**DOI:** 10.3390/ijms252011132

**Published:** 2024-10-17

**Authors:** Xiaoyu Yin, Ziwei Liu, Gegentuya Huanood, Hayate Sawatari, Keiya Shimamori, Masahiro Kuragano, Kiyotaka Tokuraku

**Affiliations:** Graduate School of Engineering, Muroran Institute of Technology, Muroran 050-8585, Japan; yinxiaoyu344@163.com (X.Y.); exokxw@163.com (Z.L.); gegentuya1996@gmail.com (G.H.); 23041040@muroran-it.ac.jp (H.S.); simauma8476@gmail.com (K.S.); gano@muroran-it.ac.jp (M.K.)

**Keywords:** amylin, amylin aggregation inhibitor, amylin fibrils, quantum dot

## Abstract

Protein aggregation is associated with various diseases caused by protein misfolding. Among them, amylin deposition is a prominent feature of type 2 diabetes. At present, the mechanism of amylin aggregation remains unclear, and this has hindered the treatment of type 2 diabetes. In this study, we analyzed the aggregation process of amylin using the quantum dot (QD) imaging method. QD fluorescence imaging revealed that in the presence of 100 μM amylin, aggregates appeared after 12 h of incubation, while a large number of aggregates formed after 24 h of incubation, with a standard deviation (SD) value of 5.435. In contrast, 50 μM amylin did not induce the formation of aggregates after 12 h of incubation, although a large number of aggregates were observed after 24 h of incubation, with an SD value of 2.883. Confocal laser microscopy observations revealed that these aggregates were deposited in three dimensions. Transmission electron microscopy revealed that amylin existed as misfolded fibrils in vitro and that QDs were uniformly bound to the amylin fibrils. In addition, using a microliter-scale high-throughput screening (MSHTS) system, we found that rosmarinic acid, a polyphenol, inhibited amylin aggregation at a half-maximal effective concentration of 852.8 μM. These results demonstrate that the MSHTS system is a powerful tool for evaluating the inhibitory activity of amylin aggregation. Our findings will contribute to the understanding of the pathogenesis of amylin-related diseases and the discovery of compounds that may be useful in the treatment and prevention of these diseases.

## 1. Introduction

Protein misfolding and aggregation are the hallmarks of many neurodegenerative diseases, including Alzheimer’s disease (AD), Parkinson’s disease, prion disease, and type 2 diabetes mellitus (T2DM) [1]. The pathological hallmarks of this class of diseases are amyloid fibrils, which are structurally conserved intracellular and extracellular insoluble proteinaceous deposits [2]. Among these diseases, T2DM is a chronic metabolic disease associated with hyperglycemia, elevated levels of glucose in plasma for prolonged periods, and insulin resistance in target tissues [3]. Currently, the incidence of T2DM has reached epidemic proportions globally and already poses a degree of risk to life [4,5]. Therefore, the treatment of T2DM is of great importance. Human islet amyloid polypeptide (hIAPP), or amylin, has been recognized as a therapeutic target for T2DM management [6].

Amylin is a 37-amino-acid peptide hormone co-secreted with insulin by pancreatic β cells. Under normal physiological conditions, the concentration of basal plasma amylin in the human body is in the range of 2–15 pmol/L, and amylin and insulin secretion occurs in a ratio of 10–100:1 (insulin/amylin) after a meal in response to various stimuli, including glucose [7]. The released amylin enters systemic circulation, crosses the blood–brain barrier, activates specific receptors in the brain, and induces satiety. It also plays a key role in inhibiting the release of glucagon from the pancreas [8,9]. However, amylin is thought to be responsible for amyloid formation in the pancreas of diabetics [10,11]. Studies have shown that amylin easily forms amyloid fibrils, which cause cell malfunction and death [12].

Studies have confirmed that the reduction in islet β cells in patients with type 2 diabetes (T2D) is mainly due to a large amount of amyloid deposition inside and outside the islet cells, and the toxic amylin oligomers increase the frequency of β-cell apoptosis beyond its regeneration frequency, with the replication of β cells being more sensitive to the toxicity of amylin oligomers, as well as preferentially apoptotic, leading to a gradual decrease in the number of β cells [13,14]. Human amylin oligomers display cytoplasmic toxicity and can induce the apoptosis of islet cells [12]. The massive apoptosis of β cells diminishes insulin secretion, leading to prediabetes or diabetes [15]. Currently, there are no cures for any protein amyloid diseases. Progress toward managing protein-misfolding diseases in general has been hampered by a failure to develop effective disease-modifying drugs [1].

In recent years, quantum dot (QD) nanoprobes have developed into very useful fluorescent probes for biological staining and diagnosis based on fluorescence 16,17,18microscopy [16,17,18]. Previously, we developed a microliter-scale high-throughput screening (MSHTS) system based on a QD imaging method for amyloid proteins [19,20]. The MSHTS system has many significant advantages. It can rapidly screen a large number of samples in a short period of time and only requires a microliter-level sample volume, saving experimental materials. In summary, the MSHTS system may become a powerful tool for screening amyloid aggregation inhibitors and thus be widely used. Rosmarinic acid (RA) is a natural polyphenol found mainly in plants. Due to its antioxidant and anti-inflammatory properties, it shows beneficial effects in the treatment of neurodegenerative diseases [21,22]. RA is a known inhibitor of Aβ aggregation, and antioxidants containing a catechol moiety have been shown to inhibit Aβ aggregation [23,24]. Therefore, in this study, we attempted to visualize the 2D and 3D aggregation of amylin by nonspecific QD binding and evaluate the aggregation inhibitory activity of amylin using the MSHTS system.

To achieve this, we used QD nanoprobes to image and analyze the aggregation process of amylin by conventional fluorescence microscopy and confocal microscopy. Thereafter, we used transmission electron microscopy (TEM) to detect the fibril structure of amylin and confirmed that QDs could bind to amylin fibrils. We used RA to perform aggregation inhibition tests on amylin. Those results further confirmed that RA had aggregation inhibition activity on amylin. Collectively, the findings of this study indicate that the MSHTS system has broad development prospects in the evaluation of amylin aggregation inhibition. In the future, through further technological innovation and wider application, it may provide strong support for screening inhibitors and treatment of amyloid-related diseases.

## 2. Results

### 2.1. The Appropriate Concentration of Sodium Acetate (SA) Buffer

To observe the aggregation and fibrillation of amylin, we first explored the concentration of the buffer to find a suitable buffer concentration. SA has been used as a buffer (pH 5.5) to study amylin [25]. Therefore, we used an acidic SA solution (pH 5.5) as the buffer for amylin aggregation.

After QDs bound to amylin, the intervening spaces between amylin aggregates became dark, allowing images of the aggregates to be detected by fluorescence microscopy. The fluorescence imaging results show that starting from 24 h of incubation, the 20 mM SA solution exhibited the most obvious aggregation (Figure 1A). In our previous study, we suggested that the standard deviation (SD) value of the fluorescence intensity of each pixel was related to the amount of amyloid aggregation [20], allowing the optimal concentration of amylin aggregation to be evaluated by the peak SD value. In this study, we compared the SD values of different concentrations of SA over time. The results show that the SD values of 20, 30, and 40 mM SA solutions reached a plateau at around 48 h. Among them, the peak SD value of the 20 mM SA solution was the highest, indicating the best aggregation effect and the shortest aggregation time (Figure 1B).

To further confirm the shape of the aggregates, we used a confocal laser scanning microscope to take a 3D image of aggregates after 120 h of incubation (Figure 2A). As time progressed, the thickness of amylin increased due to the formation of aggregates. In addition, the shapes of aggregates in 20, 30, and 40 mM SA solutions were similar. To explore the effect of different concentrations of SA on the amylin aggregation process, we also measured the thickness of 3D deposition based on all of the images of the XY view (Figure 2B). The results show that the thickness of amylin aggregates in 20, 30, and 40 mM SA solutions was similar, although the thickness of amylin aggregates in 50 mM SA solution was less than that at other concentrations. Based on the results of Figure 1 and Figure 2, the following experiments were performed in 20 mM SA solution.

### 2.2. The Appropriate Concentration of Amylin for Aggregation

Exploring the appropriate concentration of amylin that induces aggregation helps to gain a deeper understanding of the pathogenesis of diabetes. Moreover, clarifying the optimal aggregation concentration can better reveal its association with the disease process. Therefore, to observe the optimal aggregation effect of amylin, we investigated the aggregation of different concentrations of amylin. Continuous observation by conventional fluorescence microscopy revealed that with 100 μM amylin, aggregates appeared after 12 h of incubation, while a large number of aggregates were formed after 24 h of incubation. In contrast, 50 μM amylin did not form aggregates at 12 h of incubation, although a large number of aggregates were observed after 24 h of incubation (Figure 3A). We calculated the changes in the SD value of amylin aggregation over time at various concentrations. Amylin formed aggregates in a dose-dependent manner, with 100 μM amylin reaching a plateau at around 24 h (Figure 3B,C). Since there was almost no increase in the SD value below 25 µM, it was concluded that the detection limit for amylin aggregation under these conditions was around 25 µM (Figure 3C).

### 2.3. Transmission Electron Microscopy Observations of Amylin Fibrils

Next, we performed TEM to confirm whether amylin forms fibrils and whether QDs bind to them (Figure 4). TEM, which can intuitively reveal the morphology and size of fibrils, provides direct evidence for studying the formation mechanism and conformational changes of amylin. Our results show that low-magnification images indicate that amylin has misfolded fibrils in vitro. High-magnification images show that QDs are uniformly bound to amylin fibrils (Figure 4, ×20k). The deposition of these misfolded fibrils can destroy the normal structure and function of pancreatic islet cells and even cause apoptosis and death of pancreatic islet cells [26].

### 2.4. Effect of RA on the Aggregation of Amylin

We evaluated the inhibitory activity of different concentrations of RA on the aggregation of amylin using the MSHTS system. The captured 2D fluorescence images were analyzed by ImageJ software to obtain SD values. Then, an inhibition curve was drawn according to the SD values at 48 h, and the half-maximal effective concentration (EC_50_) value was calculated. The image results show that amylin aggregation was inhibited by 300 and 1500 μM RA (Figure 5A). The EC_50_ value calculated from the inhibition curve was 852.8 µM (Figure 5B). These results indicate that RA can effectively inhibit the aggregation of amylin, which may provide a new potential strategy for the treatment of T2DM. At the same time, it also proves that the MSHTS system can be used as an effective tool to evaluate the aggregation inhibitory activity of amylin proteins.

## 3. Discussion

T2D is one of the most common metabolic diseases in the world and is characterized by insulin resistance, as well as apoptosis of β cells and the formation of amyloid proteins [27]. Amylin deposition is a hallmark of T2D and is found in the pancreatic islets of more than 90% of patients with this disease [28,29]. Although there have been many advances in the study of amylin, there are still great obstacles to exploring its mechanism of action. In this study, we used the imaging method of QD nanoprobes and the MSHTS system to analyze the aggregation process of amylin and confirmed that these aggregates exhibited three-dimensional deposition by confocal laser microscopy observation.

Since proteins are susceptible to various physical and chemical degradations, ensuring their stability is important. SA is widely used as a buffer in many studies. Studies have shown that SA buffer has strong stability for proteins at acidic pH [30]. A 10 mM SA buffer was used to study the inhibition of amylin aggregation [25], while a 50 mM SA buffer was used to study amylin [31]. Our analysis of the optimal SA buffer concentration for amylin aggregation showed that 20, 30, and 40 mM SA had similar aggregation levels, slightly better than 10 mM. Three-dimensional imaging results using confocal microscopy showed that 50 mM SA had fewer amylin aggregates than other concentrations. This may be due to differences in the reaction systems and environmental conditions of different experiments, or the interaction between other reagents in the experiment and the buffer.

Abnormal aggregation of amylin is closely related to the development of T2D. To understand the pathogenesis of diabetes, we also analyzed the optimal aggregation concentration of amylin. In a study by Azzam et al., amylin was used for an aggregation inhibition assay, and the Thioflavin-T (ThT) results showed that 15 μM amylin had a high fluorescence intensity, which showed a lower amylin aggregation concentration than our fluorescence imaging results [28]. Our results showed that 50 μM amylin aggregated at 24 h, while 25 μM amylin produced almost no aggregates until 168 h (Figure 3). In our previous study, we mentioned that the difference between the MSHTS system and the ThT assay may be due to differences in detection mechanism or aggregate imaging [32].

Nowadays, TEM is a powerful tool that helps visualize the assembly mechanism and structural characteristics of amyloid [33]. To observe the fibril structure of amylin and confirm whether QDs can bind to amylin fibrils, we used TEM to detect the fibril structure of amylin. The results show that the fibril structure of amylin displayed an elongated morphology with a cross-folded structure. High-magnification images showed that QDs were uniformly bound to amylin fibrils. Our previous studies demonstrated that QDs do not alter the morphology or structure of amyloid fibrils [34]. In the high-magnification images, slight differences in the morphology of fibrils with and without QDs were observed, such as twisted and long fibers in the samples without QDs. However, in the low-magnification images, twisted and long filaments were also observed in the samples with QDs, so this difference cannot be attributed to the effect of QDs. Our results are similar to those of some studies on amylin fibrils [35,36]. The slight differences from our observations may be due to factors such as sample preparation methods (samples were placed on a copper grid for different periods of time, the staining solution and staining time are different) and observation conditions. These amylin fibrils formed in the body and deposited in organs or tissues can induce amyloidosis. Extensive amylin deposition is found in the temporal lobe gray matter and blood vessels of patients with T2D [37]. The brain tissue of diabetic patients with cerebrovascular dementia or AD contains many oligomeric amylin deposits [38]. This may indicate that excessive deposition of amylin leads not only to the occurrence of diabetes but also to the occurrence of AD. QDs are an ideal model nanostructure due to their superior optical properties that permit visual confirmation of in vivo targeting and localization and due to their potential as a bio-imaging tool [39]. Therefore, our results show that the combination of QDs with amyloid fibrils provides the possibility of more precise and sensitive positioning and characterization of fibrils. At the same time, it helps to gain a deeper understanding of the distribution, migration, and transformation process of fibrils in cells and tissues.

Currently, most oral and/or injectable medications used to treat T2DM depend on the individual’s condition and are centered around balancing blood glucose levels. Commonly used therapeutic interventions, such as metformin, inhibit glucose production in the liver, while thiazolidinediones (such as rosiglitazone) enhance peripheral insulin sensitivity. Although the use of these drugs can control T2DM, they also have certain limitations in terms of efficacy, cost, follow-up dose, and mild to severe adverse reactions [40,41]. The formation of amylin aggregates occurs much earlier than the onset of acute hyperglycemia [42]. Amyloid fibrils are very strong, stable, and difficult to decompose. Therefore, the search for amylin aggregation inhibitors is an active area of research. RA, a natural product containing catechols, has been extensively studied for its bioactivity and pharmacological properties [43]. The catechol functional group in RA can be self-oxidized to an o-quinone intermediate, which then covalently reacts with amylin to prevent the growth of amyloid protein [44]. In this study, we used RA to inhibit the aggregation of amylin based on the MSHTS system and calculated the EC_50_ values. Our data showed that RA had a high inhibitory activity against the aggregation of amylin in vitro. Our findings will help better understand the pathogenesis of T2DM caused by amylin deposition. These results suggest that the MSHTS system can be used to evaluate the in vitro inhibitory activity of amyloid aggregation. In in vitro experiments, the aggregation of amylin primarily occurs through its natural folding and aggregation mechanisms. This is similar to the abnormal folding and aggregation processes induced by various factors in the in vivo environment. This study effectively simulates the aggregation process of amylin and provides important insights into the fundamental mechanisms of aggregation.

Compared to previous studies on visualizing amylin aggregation [45,46], QDs exhibit excellent photostability, making our method of using QDs for imaging the aggregation process of amylin more stable. In addition, the combination of QDs and amylin enables real-time imaging, allowing the dynamic process of amylin aggregation to be observed continuously over a longer period compared to conventional organic fluorescent dyes that are prone to quenching. In the future, using the MSHTS system, we will probe more inhibitors that might be able to effectively prevent amylin aggregation. At the same time, we will attempt to further verify the in vivo deposition of amylin and evaluate the inhibitory activity of inhibitors through animal models.

## 4. Materials and Methods

### 4.1. Materials

Human amylin (4219-v, Peptide Institute Inc., Osaka, Japan) and QD605 (Q21501MP, Thermo Fisher Scientific, Waltham, MA, USA) were purchased commercially.

### 4.2. Preparation of Amylin

Five mg of amylin (1–37, human, amide 5 mg) was dissolved in 5 mL of HFIP (1,1,1,3,3,3-hexafluoro-2-proranol, CAS No. 920-66-1). The bottle was wrapped with Parafilm for 1 h (agitating gently for 1 min every 10 min at room temperature). At 25 °C, the solution was ultrasonically shaken for 10 min using an ultrasonic cleaner (50 Hz, FU-2H, Tokyo Glass Kikai Co., Ltd., Tokyo, Japan). The dissolved amylin was aliquoted into 1.5 mL tubes, at 19.5 μL (5 nmol) per tube. Tubes were placed on a clean bench for 24 h, with the extractor fan turned on, to completely volatilize HFIP. After the visual confirmation of complete volatilization, the mouth of each tube was covered with Parafilm, and the tubes were stored at −80 °C. SA (pH 5.5) was used as the buffer.

### 4.3. MSHTS System

The EC_50_ values of inhibitors were determined by a modified MSHTS system, as described in our previous report [32]. The image was captured with an inverted fluorescence microscope (TE2000, Nikon, Tokyo, Japan) using a 4× objective equipped with a color CCD camera (DP72, Olympus, Tokyo, Japan). Confocal laser microscope aggregates in a 1536-well plate (782096, Greiner, Kremsmünster, Austria) were observed by a confocal laser microscope (Nikon C2 Plus, Nikon, Tokyo, Japan) using a 20× objective. Amyloid protein aggregation was visualized under a fluorescence microscope, and QD605 and proteins were dispersed in samples, showing a red color. The SD values of the fluorescence intensity of 40,000 pixels (200 × 200 pixels) around the central area of each well were measured by ImageJ software version 1.53b (National Institutes of Health, Bethesda, MD, USA). EC_50_ was estimated from the SD values by Prism software (6.01, GraphPad software, San Diego, CA, USA) using an EC_50_ shift by global fitting (asymmetric sigmoidal, five-parameter logistic).

### 4.4. The Appropriate Concentration of SA Buffer for Amylin Aggregation

The appropriate buffer concentration used was based on data in a related paper [25]. We first prepared 20, 40, 60, 80, and 100 mM SA solution samples and 100 μM amylin (containing 100 nM QDs). We thawed 1 tube of amylin (5 nmol) and added 49.38 μL ultrapure water and 0.62 μL QD (8 μM) to obtain a 100 μM amylin solution. Each SA sample was evenly mixed with amylin solution at a 1:1 ratio and centrifuged (10,000× *g*, 4 °C, 2 min). Then, 5 μL of supernatant was injected into a 1536-well plate and centrifuged at 3700 rpm for 5 min at room temperature. After centrifugation, samples were incubated at 37 °C. Aggregation images were observed and captured using an inverted fluorescence microscope and a confocal laser microscope. ImageJ software was used to analyze the SD values and plot the time-dependent curves of each concentration of SA.

### 4.5. The Appropriate Concentration of Amylin for Aggregation

To a thawed tube of amylin, SA was added to make a 200 μM amylin solution. Amylin was then diluted to 12.5, 25, 50, and 100 μM. After all samples were centrifuged (10,000× *g*, 4 °C, 2 min), 5 μL of supernatant from each sample was injected into a 1536-well plate and centrifuged at 3700 rpm for 5 min at room temperature. Aggregation images were observed and captured using an inverted fluorescence microscope. ImageJ software was used to analyze the SD values and plot the time-dependent curves of each concentration of amylin.

### 4.6. TEM Observation

Amylin samples were prepared in a total of 100 μL of 20 mM SA and incubated in 1.5 mL tubes at 37 °C. Samples were then deposited in 10 μL aliquots onto 200-mesh copper grids for 5 min, dried with filter paper, and then washed twice with 1 × PBS. After washing with 1 × PBS, samples were negatively stained twice with 1% phosphotungstic acid and dried on filter paper. Specimens were examined under an H-7600 transmission electron microscope (Hitachi, Tokyo, Japan) at 8k and 20k magnifications.

### 4.7. Inhibition of Amylin Aggregation by RA

RA was dissolved in 99.5% ethanol to prepare 0.96–3000 μM RA solutions. Separately, 100 μM amylin (containing 100 nM QDs) was prepared. We thawed 1 tube of amylin (5 nmol) and added 49.38 μL SA (40 mM) and 0.62 μL QD (8 μM) to obtain a 100 μM amylin solution. Amylin and various concentrations of RA solution were mixed at a 1:1 ratio and centrifuged (10,000× *g*, 4 °C, 2 min). Then, 5 μL of supernatant was injected into a 1536-well plate and centrifuged at 3700 rpm for 5 min at room temperature. After centrifugation, samples were incubated at 37 °C. Images of the aggregation were observed and captured with a color CCD camera. The captured fluorescence images were analyzed by ImageJ software to obtain the SD values. The EC_50_ value was calculated based on an inhibition curve.

## 5. Conclusions

In summary, we successfully visualized amylin aggregation using QD fluorescence imaging and detected the fibril structure of amylin by TEM. In addition, we evaluated the aggregation inhibitory activity of RA in amylin with the MSHTS method using nonspecific QD binding. Our results help to reveal the pathogenesis of amylin-related diseases. Additionally, screening for compounds that can inhibit amylin aggregation may offer valuable insights for the prevention and mitigation of these conditions. In the future, this imaging method can be applied to visualize amyloid aggregation and screen inhibitors. At the same time, we will attempt to study the aggregation and inhibition of amylin in vivo and use the MSHTS system for screening a wider range of inhibitors.

## Figures and Tables

**Figure 1 ijms-25-11132-f001:**
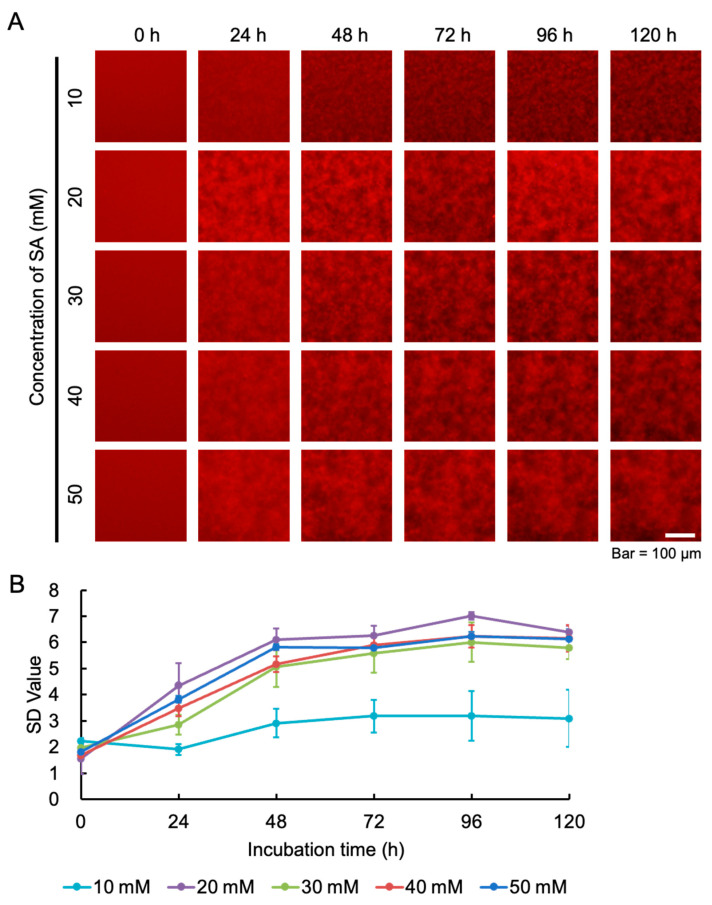
Time-lapse fluorescence imaging of aggregation processes of 50 μM amylin using 50 nM QD605 in 10–50 mM buffer (pH 5.5) conditions. (**A**) Images were captured every 24 h. The fluorescence images were trimmed to 200 × 200 pixels. (**B**) The SD values of amylin aggregation under different concentrations of SA were determined by ImageJ software. Data represent the means from three independent samples.

**Figure 2 ijms-25-11132-f002:**
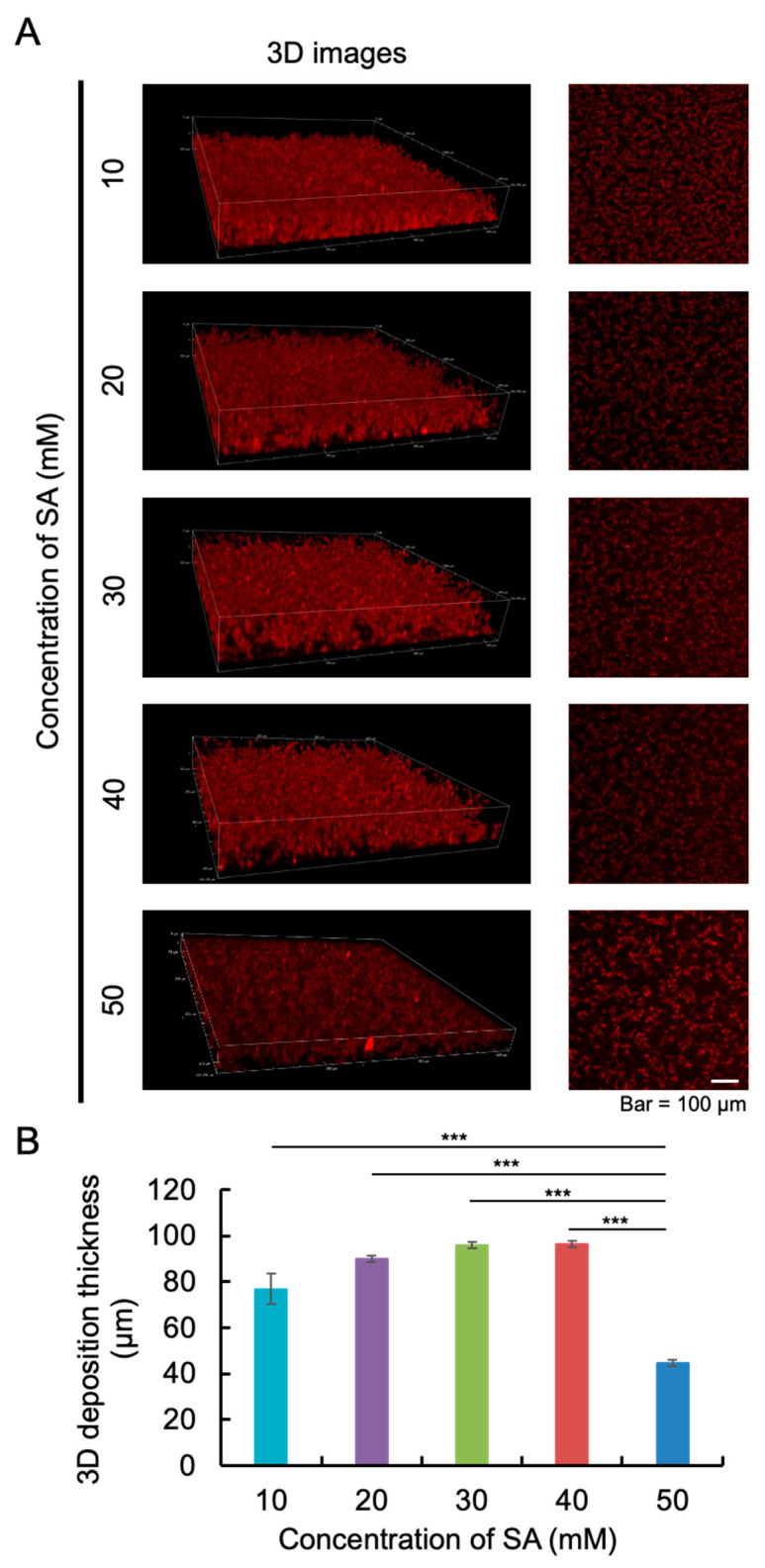
Three-dimensional imaging of amylin aggregates in SA buffer at different concentrations. (**A**) Aggregation of 50 μM amylin in 10–50 mM SA buffer containing 50 nM QD605 was imaged in 3D by confocal laser microscopy. Three-dimensional images of amylin after 120 h of incubation are represented on the left, along with slice images of amylin aggregates (on the right) at each SA concentration. (**B**) Aggregation thickness of amylin in 10–50 mM SA buffer was measured. Data represent the means from three independent samples (***: *p* < 0.001, Welch’s *t*-test).

**Figure 3 ijms-25-11132-f003:**
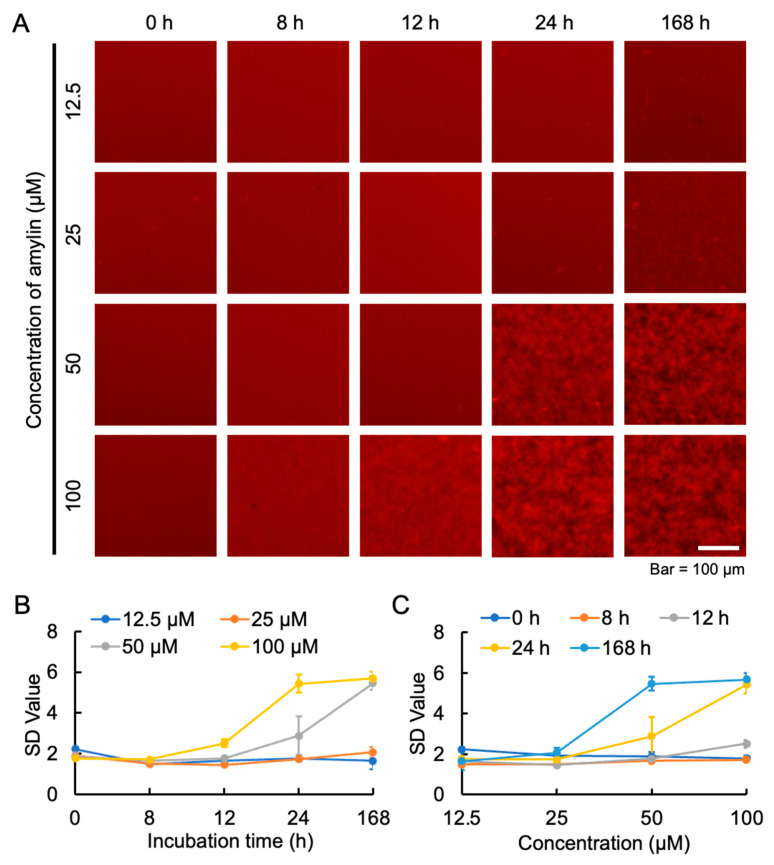
Time-lapse fluorescence imaging of aggregation processes of 12.5–100 μM amylin using QD605. (**A**) Images of amylin aggregation were collected at 8, 12, 24, and 168 h. The fluorescence images were trimmed to 200 × 200 pixels. (**B**) SD values of amylin concentrations over time were determined using ImageJ software. (**C**) SD values of amylin concentration increase at different time points were determined using ImageJ software. Data represent the means from three independent samples.

**Figure 4 ijms-25-11132-f004:**
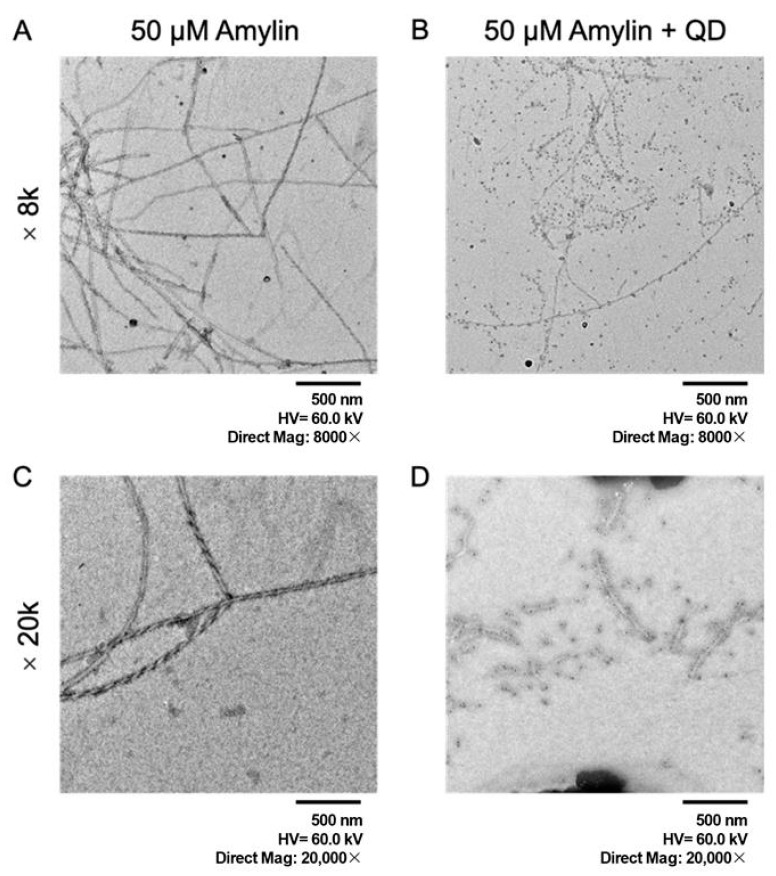
TEM observations of amylin fibrils with QD nanoprobes under various conditions: (**A**) without QDs at 8k magnification. (**B**) with QDs at 8k magnification. (**C**) without QDs at 20k magnification. (**D**) with QDs at 20k magnification.

**Figure 5 ijms-25-11132-f005:**
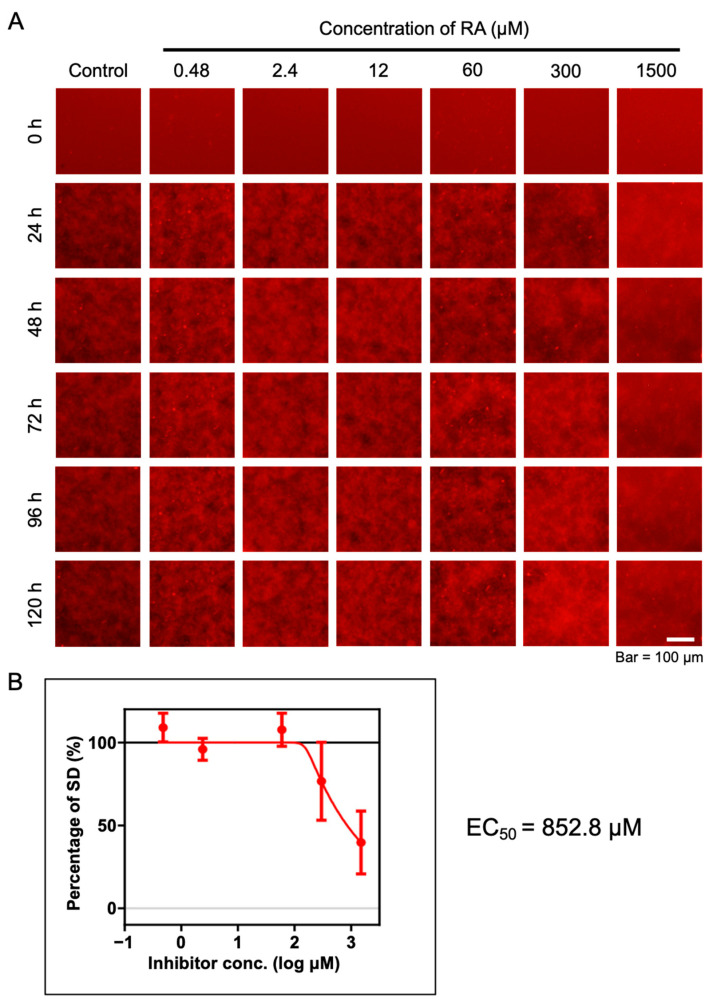
Estimation of EC_50_ by the MSHTS system. (**A**) Fluorescence image of 50 μM amylin protein treated by 0.48–1500 μM RA. (**B**) Aggregation inhibitory activity of amylin by 0.48–1500 μM RA. The SD values of the amylin aggregation fluorescence images at 48 h were measured. The inhibition curve was drawn from SD values, and the EC_50_ value was calculated based on the inhibition curve. Data represent the means from three independent samples.

## Data Availability

The original contributions presented in the study are included in the article, further inquiries can be directed to the corresponding author.

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
