# Peer review of "Analyzing Amylin Aggregation Inhibition Through Quantum Dot Fluorescence Imaging"

_ijms, 2024, doi:10.3390/ijms252011132_

Round 1
Reviewer 1 Report
Comments and Suggestions for Authors
This study used quantum-dot imaging to track amylin aggregation, showing significant aggregation at 100 μM after 12 hours and at 50 μM after 24 hours. Confocal and electron microscopy confirmed three-dimensional deposits and misfolded fibrils. Rosmarinic acid inhibited aggregation with an EC50 of 852.8 μM, highlighting a potential screening tool for treatments targeting protein misfolding diseases. Overall, this work is well-prepared and can be considered with several major revisions.
1. The imaging of aggregation processes can be illustrated with more proper words than “Real-time”as the imaging intervals are ~ 24 hrs.
2. The reason for quantifying amylin aggregates in the 3D images using the thickness should be discussed. The N number should be given in Fig2.
3. The figure formats should be consistent. The text size in Fig4 is different from other figures.
4. It seems that the incubation of QDs alters the amylin fibrils morphology and structures. Detailed discussions should be provided.
5. The significance and advancements of this work should be highlighted in the abstract and conclusions.
6. To what extent the experiment mimic real in vivo pathological processes of amylin aggregation should be discussed.
7. A comparison of this study with previous work for visualizing amylin aggregation may be provided to illustrate the novelty of this study.
Comments on the Quality of English LanguageModerate editing of English language required.
Author Response
Comments and Suggestions for Authors:
This study used quantum-dot imaging to track amylin aggregation, showing significant aggregation at 100 μM after 12 hours and at 50 μM after 24 hours. Confocal and electron microscopy confirmed three-dimensional deposits and misfolded fibrils. Rosmarinic acid inhibited aggregation with an EC50 of 852.8 μM, highlighting a potential screening tool for treatments targeting protein misfolding diseases. Overall, this work is well-prepared and can be considered with several major revisions.
Response:
Thank you for your positive evaluation. We have carefully considered your comments and made revisions where necessary.
Comment 1:
- The imaging of aggregation processes can be illustrated with more proper words than “Real-time” as the imaging intervals are ~ 24 hrs.
Response 1:
Thank you for pointing this out. We have changed "real-time imaging" to "fluorescence imaging" in the manuscript. In some cases, we represented the observations over time as “time-lapse fluorescence imaging”.
Comment 2:
- The reason for quantifying amylin aggregates in the 3D images using the thickness should be discussed. The N number should be given in Fig2.
Response 2:
Following reviewer comments, we added the reason to measure aggregate thickness by 3D imaging (lines 111-112). The 3D deposition thickness can reflect the approximate amount of amylin aggregates. By quantifying the deposition thickness, we can clearly know the effect of different concentrations of sodium acetate on the amylin aggregation process. We have also added the N number in Figure 2 according to the reviewer's suggestion.
Comment 3:
- The figure formats should be consistent. The text size in Fig4 is different from other figures.
Response 3:
Thank you for pointing this out. We have modified the text size in Figure 4.
Comment 4:
- It seems that the incubation of QDs alters the amylin fibrils morphology and structures. Detailed discussions should be provided.
Response 4:
Thank you for pointing this out. We have added the discussion about fibril morphology to lines 210-215 of the manuscript. The focus of our TEM observation in this study is to compare between the presence and absence of QDs and to show the binding of QDs to amylin fibrils. Consequently, the images were primarily focused on the QDs. As the reviewer commented, from the high-magnification TEM images alone, it appears that QDs affect the morphology of amylin fibrils. However, when the low-magnification images are taken together, we believe that it cannot be said that QDs affect fibril morphology. Of course, we cannot completely rule out the possibility that QDs may affect the morphology of amylin fibrils, but this will require detailed structural studies using cryo-electron microscopy, solid-state NMR, etc., which is a future challenge.
Comment 5:
- The significance and advancements of this work should be highlighted in the abstract and conclusions.
Response 5:
Thank you for pointing this out. We have added the significance and advancements of this study in the abstract (lines 24-26) and conclusions (330-332).
Comment 6:
- To what extent the experiment mimic real in vivo pathological processes of amylin aggregation should be discussed.
Response 6:
Thank you for pointing this out. We have added the discussion to lines 248-252 of the manuscript.
Comment 7:
- A comparison of this study with previous work for visualizing amylin aggregation may be provided to illustrate the novelty of this study.
Response 7:
We have added the discussion to lines 253-258 of the manuscript according to the reviewer’s comment.
Reviewer 2 Report
Comments and Suggestions for Authors
In this manuscript, authors mainly used QD to image amylin aggregations and evaluated the QD-binded amylin aggregation with TEM and confocal laser microscopy. This study also evaluated the aggregation inhibitory activity of rosmarinic acid (RA) in amylin with the MSHTS method using nonspecific QD binding. The structure and strategy of this manuscript are highly similar to author's previous paper (amyloid aggregations). I suggest the topic of this manuscript could be modified, because current topic is almost the same in previous paper (Int. J. Mol. Sci. 2020, 21(6), 1978; https://doi.org/10.3390/ijms21061978). Some questions and comments for this manuscript are as below.
1. Based on experimental results, QDs certainly could bind to amylin aggregations. What interaction is between QD and amylin aggregations?
2. The interaction seems to use nonspecific QD binding. This is a very important issue that how QDs could recognize amylin aggregations when other biomolecules exist in the same microenvironment. In this study, authors only used amylin-included system to test and QDs bind to amylin aggregations with the nonspecific binding. This means it is hard to apply in real cases. I suggest that authors should use specific binding to develop this system. Otherwise, current experimental data has no meaning.
3. Why did authors use the QD concentration of 100 nM? Is 100 nM the best concentration in this manuscript?
4. Based on figure 4 (TEM images), do QDs have the inhibition effect of amylin aggregations? Because QDs seems to interrupt the growth of amylin fibers.
5. Please provide detail experimental parameters in Figure 4.
6. Using SD as a standard to identify amylin aggregations is only suitable in this manuscript. When other biomolecules exist in the same system, SD is not a suitable standard. If using fluorescent intensity of QDs will be better than SD? And many parameters can affect the SD in a real situation.
7. Please describe and give the appropriate concentration of SA buffer for amylin aggregation.
8. I suggest that the detection limit of amylin aggregations could be showed.
9. If using QDs to image amylin aggregations is better than using ThT? Authors could add this to discuss.
Comments on the Quality of English Language
The quality of English is ok.
Author Response
Comments and Suggestions for Authors:
In this manuscript, authors mainly used QD to image amylin aggregations and evaluated the QD-binded amylin aggregation with TEM and confocal laser microscopy. This study also evaluated the aggregation inhibitory activity of rosmarinic acid (RA) in amylin with the MSHTS method using nonspecific QD binding. The structure and strategy of this manuscript are highly similar to author’s previous paper (amyloid aggregations). I suggest the topic of this manuscript could be modified, because current topic is almost the same in previous paper (Int. J. Mol. Sci. 2020, 21(6), 1978; https://doi.org/10.3390/ijms21061978). Some questions and comments for this manuscript are as below.
Response:
Thank you very much for your feedback. According to your suggestion, I have modified the topic of the manuscript. I have responded to your comments and suggestions below.
Comment 1:
- Based on experimental results, QDs certainly could bind to amylin aggregations. What interaction is between QD and amylin aggregations?
Response 1:
In our previous studies, we adopted polyethylene glycol (PEG)-conjugated amino Qdot (QD-PEG-NH2). Although our previous reports have shown that QD-PEG-NH2 binds to amyloid aggregates such as Aβ, Tau, α-synuclein, and SAA, the binding mechanism remains unknown. We speculate that the mechanism is as follows: The polar functional groups in amylin facilitate the formation of hydrogen bonds with the amino groups, thereby enhancing the specificity of the binding. Additionally, the unique three-dimensional structure of amylin allows it to fit effectively on the surface of the amino-functionalized QDs, promoting a stable interaction.
Comment 2:
- The interaction seems to use nonspecific QD binding. This is a very important issue that how QDs could recognize amylin aggregations when other biomolecules exist in the same microenvironment. In this study, authors only used amylin-included system to test and QDs bind to amylin aggregations with the nonspecific binding. This means it is hard to apply in real cases. I suggest that authors should use specific binding to develop this system. Otherwise, current experimental data has no meaning.
Response 2:
As the reviewer points out, we also believe that observations using the nonspecific binding of QDs are not suitable under conditions such as in vivo, where other biomolecules coexist. On the other hand, our previous work (see references in the manuscript) revealed that the QDs used in our experiments nonspecifically bind not only to amylin but also to many other amyloid aggregates, including Aβ, tau, α-synuclein, and SAA. This made it possible to compare and verify the formation processes of various amyloid aggregates under similar conditions. We believe that analysis using nonspecific binding of QDs will be useful for various in vitro studies that do not involve diverse biomolecules, such as the evaluation of the aggregation process and physical properties of aggregates, and screening of candidate aggregation inhibitors. As the reviewer commented, the study of amylin aggregation in vivo will require the use of specific probes such as QD-Aβ, which we are currently investigating, and this is our next challenge.
Comment 3:
- Why did authors use the QD concentration of 100 nM? Is 100 nM the best concentration in this manuscript?
Response 3:
Thank you for pointing this out.
In the methods section, we refer to "containing 100 nM QDs" within a 100 μM amylin solution. Following the mixing with sodium acetate (method 4.4) or rosmarinic acid (method 4.7) in a 1:1 ratio, the resulting final concentration of amylin and QDs were 50 µM and 50 nM, respectively. The addition of 0.1% QDs relative to the protein concentration was the concentration optimized in our previous study (References 18 and 20). We also added these final concentrations to the legends in Figures 1 and 2.
Comment 4:
- Based on figure 4 (TEM images), do QDs have the inhibition effect of amylin aggregations? Because QDs seems to interrupt the growth of amylin fibers.
Response 4:
Thank you for pointing this out. We have added the discussion about TEM observation to lines 210-215 of the manuscript. As the reviewer commented, from the high-magnification TEM images alone, it appears that QDs affect fibril length. However, because long filaments are also visible in the low-magnification images, we believe that it is not possible to say that QDs inhibit aggregation.
Comment 5:
- Please provide detail experimental parameters in Figure 4.
Response 5:
Thank you for pointing this out. We have enlarged the experimental parameters in Figure 4 for the reader's reading.
Comment 6:
- Using SD as a standard to identify amylin aggregations is only suitable in this manuscript. When other biomolecules exist in the same system, SD is not a suitable standard. If using fluorescent intensity of QDs will be better than SD? And many parameters can affect the SD in a real situation.
Response 6:
Thank you for pointing this out.
Since the total amount of QDs in the sample solution is the same, amylin aggregation does not result in a change in total fluorescence intensity. Following the binding of QDs to amylin, a difference in brightness emerges among the amylin aggregates. This brightness difference is quantified using the SD value measured by ImageJ to assess the degree of aggregation. Therefore, using the SD for statistical analysis in this study is more appropriate.
Comment 7:
- Please describe and give the appropriate concentration of SA buffer for amylin aggregation.
Response 7:
Thank you for pointing this out.
We mentioned in lines 99-101 that a concentration of 20 mM SA is appropriate for amylin aggregation.
Comment 8:
- I suggest that the detection limit of amylin aggregations could be showed.
Response 8:
Following a reviewer's suggestion, we have added a statement on lines 138-140 regarding the detection limit of amylin aggregation under these conditions.
Comment 9:
- If using QDs to image amylin aggregations is better than using ThT? Authors could add this to discuss.
Response 9:
Thank you for pointing this out. We agree with this comment. We have added the discussion to lines 253-258 of the manuscript. The photostability of QDs and their ability to enable real-time imaging of amylin is superior to that of ThT. Therefore, the use of QDs allows the dynamic process of amylin aggregation to be continuously observed over a longer period.
Round 2
Reviewer 1 Report
Comments and Suggestions for Authors
Authors have addressed all of my concerns.
Comments on the Quality of English LanguageMinor editing of English language required